# Study on the Properties of Transparent Bamboo Prepared by Epoxy Resin Impregnation

**DOI:** 10.3390/polym12040863

**Published:** 2020-04-09

**Authors:** Yan Wu, Yajing Wang, Feng Yang, Jing Wang, Xuehua Wang

**Affiliations:** 1College of Furnishings and Industrial Design, Nanjing Forestry University, Nanjing 210037, China; lionzaka@163.com (Y.W.); WangJing_9711@163.com (J.W.); xuehua3099@sina.com (X.W.); 2Co-Innovation Center of Efficient Processing and Utilization of Forest Resources, Nanjing Forestry University, Nanjing 210037, China; 3Fashion Accessory Art and Engineering College, Beijing Institute of Fashion Technology, Beijing 100029, China

**Keywords:** transparent bamboo, epoxy resin, heat treatment, acid delignification, light transmittance

## Abstract

In this paper, Moso bamboo (*Phyllostachys heterocycle*) before and after heat treatment were used as raw materials to prepare transparent bamboo (TB). In an acidic environment, the lignin contained in the bamboo material was removed to obtain a bamboo template, and an epoxy resin similar to the cellulose refractive index was used for vacuum impregnation into the bamboo template to obtain a transparent bamboo material. The purpose of this study was to compare the physical and chemical properties of TB and original bamboo and the differences between TBs before and after heat treatment, taken from different parts of bamboo, in order to explore the performance advantages and disadvantages of TB as a new material. The Fourier transform infrared spectroscopy analysis (FTIR), scanning electron microscope testing (SEM), three elements analysis, light transmittance testing, and mechanical strength testing were used to study the molecular composition, microstructure, chemical composition, light transmittance, and tensile strength of the TB samples. The results showed that the lignin content of the delignified bamboo templates was greatly reduced. In addition, the SEM images showed that a large amount of epoxy resin (type E51 and type B210 curing agent) was covered on the cross-section surface and pores of the TB samples. The FTIR showed that the epoxy molecular groups appeared on the TB, and the delignified bamboo template and the resin had a good synergy effect. According to the light transmittance testing, the original bamboo samples hardly contained light transmittance under visible light. The transmittance of transparent inner bamboo (TIB) and transparent heat-treated inner bamboo (THIB) could reach about 11%, and the transmittance of transparent outer bamboo (TOB) and transparent heat-treated outer bamboo (THOB) was about 2%. The light transmittance had been significantly improved when compared with the original bamboo samples. The transmittances of the TB samples before and after heat treatment in different parts of bamboo were different. In the visible light irradiation range, the light transmittances of TB samples were as follows: TIB > THIB and THOB > TOB. Meanwhile, the tensile strength of TB was reduced, especially for TOB and THOB. In addition, TB has a wide range of raw materials, and the preparation process is environmentally friendly. It can be used for decorative materials in homes, buildings, etc., and has a great application potential.

## 1. Introduction

As people pay more attention to the environment, more and more natural materials are used in industrial production. Among many natural materials, bamboo has a short growth cycle of 3–5 years, while wood, as one of the traditional industrial materials, has a growth cycle of 20–60 years [1]. Therefore, compared with wood, bamboo has obvious environmental advantages. Globally, there are many types and wide distributions of bamboo, with about 1500 bamboo species and 36 million hectares of planting area spreading across Asia, America, and Africa [2]. In addition, bamboo has good physical and chemical properties, and its mechanical properties are excellent. The tensile strength of the down grain is about twice that of wood of the same density [3]. Furthermore, it has good abrasion resistance, water resistance, insect resistance, and mildew resistance, so it is widely used among various fields [4,5,6]. In recent years, bamboo modification has become a research hotspot for scholars at home and abroad. After a certain chemical modification of bamboo, it can become a composite material with low cost, high efficiency, and high performance [7,8] such as bamboo plastic composite material [9], bamboo fiber green composite material [10], etc.

The transparency of bamboo is related to the chemical composition and structure of bamboo. From the perspective of chemical composition, bamboo and wood contain the same main chemical components. The higher chemical components in bamboo are cellulose, hemicellulose, and lignin, accounting for more than 80% [11]. Among them, cellulose and hemicellulose are colorless substances, lignin is a colored substance, and there are other substances with different refractive indices. These substances cause the material to undergo strong light scattering and light absorption under visible light irradiation, thereby displaying a certain opaque color [12]. At present, transparent wood has achieved certain results in theory and experiments. In 1992, Fink et al. [13] pioneered the concept of transparent wood in theory. In 2016, Zhu et al. [14] first used transparent wood impregnated with epoxy resin to prepare transparent wood, and experimentally demonstrated the feasibility of preparing transparent wood. The treatment for preparing a delignified wood template is to cook a wood sample in an aqueous solution of sodium hydroxide (NaOH) and sodium sulfite (Na_2_CO_3_) for 12 h, and then place the sample in a boiling hydrogen peroxide (H_2_O_2_) bleach solution to cook to white. Transparent wood with a light transmittance as high as 79.4%, but a haze as high as 88.7%, can be prepared. In the same year, Li et al. [15] used a sodium hypochlorite (NaClO_2_) delignification method to prepare a delignified wood template in an acidic environment and then impregnated polymethyl methacrylate (PMMA) to prepare a transparent wood with a transparency of 91.7% and a haze of 48.9%. In 2017, Yaddanapudi et al. [16] used sodium chlorite to remove lignin at 95 °C for 12 h, and vacuum impregnated PMMA in the delignified wood templates. They obtained transparent wood with a light transmission of up to 70% and a haze of 46%. In addition, they tested the mechanical properties of transparent wood and found that the mechanical properties of transparent wood were better than those of the original wood, so they proposed that transparent wood could be used in the next generation of intelligent building. In 2019, Jia et al. [17] used a method of impregnating epoxy resin after delignification of sodium hypochlorite to prepare highly transparent wood with a light transmittance of up to 90% and a haze of only 10%, indicating that transparent wood can become a new energy-saving material to replace glass and other substances, and has good market prospects.

From the perspective of structure, the structure of bamboo itself is significantly different from that of wood [18]. Bamboo is a monocotyledonous plant without ray cells, which contain dense vascular bundle tissue and colloidal substances [19,20,21]. Therefore, when it is chemically modified, it is often difficult for the treating agent to penetrate into the material, and the expected effect of the modification cannot be achieved.

At present, the research of transparent wood has made some progress, and its good optical properties has made it a new material to replace glass and other compounds in the field of architecture and the home. Bamboo is also an excellent natural material, which has more obvious environmental advantages and mechanical properties than wood. Therefore, we considered improving the light transmittance of bamboo to prepare transparent bamboo to better replace the application of glass in the industrial field.

In consideration of the relevant characteristics, we selected the samples before and after heat treatment in different parts of bamboo as raw materials. The samples were immersed in an aqueous sodium hypochlorite solution, and at the same time, the bamboo samples were fully immersed in the solution by microwave treatment, and delignified bamboo templates were obtained in an acidic environment. Then, the transparent bamboo samples could be obtained by vacuum impregnating the epoxy resin into the delignified bamboo template samples. In addition, the physical and chemical properties of different transparent bamboo were compared.

## 2. Experimental Part

### 2.1. Materials

Moso bamboo (*Phyllostachys heterocycla*), 4-year-old, was produced in Anji, Zhejiang, China. It was drawn after longitudinal scrutiny, and two basic units of inner bamboo (IB) and outer bamboo (OB) were taken. Part of the IB and OB samples were heat-treated to become heat-treated inner bamboo (HIB) and heat-treated outer bamboo (HOB). The heat treatment time was 2 h, the pressure was 0.2 MPa, and the temperature was 110 °C. The equipment used for the heat treatment was an electric constant temperature air-blast drying oven, which was provided by Xinmiao Medical Equipment Manufacturing Co. Ltd., Shanghai, China. IB, OB, HIB, HOB were used as raw materials, which were sealed and refrigerated for the subsequent experiment. The specifications of the raw materials and the heat-treated materials are shown in Table 1.

In the above table, L represents length, b represents width, h represents thickness, and SD represents standard deviation. The air dry mass is M_1_ and the absolute dry mass is M_2_. The absolute water content is W_1_, and the relative water content is W_2_. According to W_1_ = (M_1_ − M_2_)/M_1_ and W_2_ = (M_1_ − M_2_)/M_2_, the absolute moisture content and relative moisture content of the materials can be obtained. The results show that the absolute moisture contents of IB, OB, HIB and HOB were 9.67%, 10.67%, 4.84%, and 4.00% and the relative moisture contents were 8.82%, 9.64%, 4.62%, and 3.85%, respectively. 

The sodium chlorite (NaClO_2_) was provided by Macklin Co. Ltd., Shanghai, China. The acetic acid (CH_3_COOH) was provided by Nanjing Chemical Reagent Co., Ltd. Nanjing, China. The absolute ethanol (99.5 %) was provided by Sinopharm Chemical Reagent Co., Ltd. Shanghai, China. The epoxy resin (type E51, epoxy value 0.48–0.54) and its curing agent (type B210, amine value 300–400 KOH mg/g) were obtained from Kunshan Julimei Electronic Materials Co., Ltd. Kunshan, China.

### 2.2. Experimental Method

The preparation process is shown in Figure 1.

#### 2.2.1. Pre-Processing

The raw bamboo materials were dried in an oven at 103 ± 2 °C for 12 h to an absolute dry state, and stored in absolute ethanol.

#### 2.2.2. Preparation of Bamboo Templates

First, 4 wt% sodium chlorite was mixed with water. After fully stirring, glacial acetic acid was added dropwise to make the solution with a pH value of 4.6. Then, the dried bamboo samples were put into the solution, a digital display three constant temperature water tank was used to heat it at 80–90 °C for 2–4 h, and then the microwave operation was conducted. The power of the microwave was 800 W and the treatment time was 5–15 s. Then, the samples continued to heat in a constant-temperature water tank for 2–3 h and restored in anhydrous ethanol for 24 h. After that, the delignified bamboo templates were obtained including delignified inner bamboo (DIB), delignified outer bamboo (DOB), delignified heat-treated inner bamboo (DHIB), and delignified heat-treated outer bamboo (DHOB).

#### 2.2.3. Fabrication of Transparent Bamboo

The bamboo template samples were taken out from the absolute ethanol and dried in a vacuum kettle for 20 min. The epoxy resin impregnation solution was prepared with an epoxy resin to curing agent ratio of 2:1. Then, the dried bamboo template samples were immersed in the epoxy resin solution for 30 min under vacuum. The impregnated samples were then placed in the silica gel sheet and cured at room temperature for more than 12 h to obtain TB samples.

### 2.3. Characterization

#### 2.3.1. Fourier Transform Infrared Spectrum (FTIR) Analysis

The samples were ground into powders and mixed with potassium bromide (KBr), then ground again and placed in a manual tablet press for tableting. Next, the pressed transparent sample sheets were put into the Vertex 80V infrared spectrum analyzer (Germany Bruker Co. Ltd., Karlsruhe, Germany) in order to analyze the changes in the molecular groups before and after treatment.

#### 2.3.2. Scanning Electron Microscope (SEM) Testing

The ultra-thin slicer was used to cut the samples from the lateral direction and sprayed them by gold atom under vacuum. The morphology of the sample cross section was magnified 200 times and 800 times by the Quanta 200 scanning electron microscopy (FEI Company, Hillsboro, OR, USA) at 3 kV voltage, respectively.

#### 2.3.3. Lignin Content Testing

This test explores the effects of the delignification process used in this study on the transparent bamboo by testing three chemical elements (lignin, cellulose, and hemicellulose) in bamboo. During the experiment, the National Renewable Energy Laboratory (NREL) method was used to measure and compare the content of lignin in IB, OB, HIB, HOB, and bamboo template samples made of the above raw materials by taking two parallel samples from each sample [22].

#### 2.3.4. Light Transmittance Testing

The Lambda 950 ultraviolet-visible spectrophotometer (Perkinelmer Co. Ltd., Waltham, MA, USA) was used to test the transmittance of IB, OB, HIB, HOB, and TB samples prepared using the above raw materials at a wavelength from 350 nm to 800 nm. Two repeats of select samples under the same experimental conditions were tested to reduce the experimental errors.

#### 2.3.5. Mechanical Strength Testing

The mechanical tensile properties of IB, OB, HIB, HOB, the delignified bamboo templates, and TB samples prepared using the above product types were measured using an AG-IC precision electronic mechanics experimental machine (Shimadzu Corporation, Kyoto, Japan). Three sets of parallel samples of each sample were tested to obtain the average value. This testing was performed at room temperature and atmospheric pressure. The fixture of the testing machine stretched the sample along the longitudinal direction of the bamboo until the longitudinal fiber in the middle of the samples was broke. The stretching speed was set to 5 mm/min and the maximum load force to 10,000 N [23,24].

## 3. Results and Discussion

### 3.1. FTIR Analysis

Figure 2 presents the FTIR curves of IB and HIB before and after delignification and vacuum impregnation. The characteristic absorption peaks of the IB sample include the stretching vibration of the O–H group at 3413 cm^−1^, the stretching vibration of the C–H group at 2919 cm^−1^ [25], the stretching vibration of acetyl groups in hemicellulose at 1737 cm^−1^ [26], the stretching vibration of lignin aromatic ring groups at 1617 cm^−1^, the stretching vibration of C=O groups at 1513 cm^−1^, and the symmetric stretching vibration of the C=O group at 1460 cm^−1^. The intensity of the absorption peak at 1617 cm^−1^ of the inner bamboo sample after heat treatment (HIB) was obviously weakened, and the others did not change significantly. This shows that the HIB sample is not significantly different from the IB sample at the molecular level. Curves 3 and 4 in Figure 2 represent the delignified inner bamboo (DIB) and delignified heat-treated inner bamboo (DHIB) samples. It can be seen that the absorption peaks of the DIB and DHIB samples disappeared at 1617 cm^−1^, 1513 cm^−1^, and 1460 cm^−1^. This means that the lignin in IB and HIB is largely removed after the delignification method, which was mentioned in Section 2.2.2, Preparation of bamboo templates [27]. Curves 5 and 6 in Figure 2 represent the transparent inner bamboo (TIB) and transparent heat-treated inner bamboo (THIB) samples. It can be seen that the infrared spectrum of TIB and THIB had newly appeared absorption peaks on the epoxy resin including 1510 cm^−1^ due to para-substituted benzene ring bending vibration at –C=C–, antisymmetric stretching vibration due to fatty aromatic ether C–O–C at 1247 cm^−1^, and out-of-plane deformation of para-substituted benzene ring =CH at 830 cm^−1^.

It can be seen from Figure 3 that the FTIR of the OB and HOB before and after delignification and vacuum impregnation was basically the same as that of the IB and HIB. The slight difference was that the OB sample did not show an absorption peak at 1617 cm^−1^. There was no significant difference between the HOB and OB samples, indicating that the heat treatment had less impact on OB than on IB.

These tests showed that there was no significant change in the internal molecules of the sample after heat treatment and the internal lignin of the sample after delignification was removed. The TB sample contained not only the molecular groups of the bamboo itself, but also the molecules of the epoxy resin.

### 3.2. SEM Testing

It can be seen from Figure 4a that there was a vascular bundle tissue in the cross section of IB, the thin-walled cells were intact, and the cell cavity contained starch. According to Figure 4b, the vascular bundle tissues and parenchyma cells in the IB were not significantly damaged after heat treatment, but the starch material in the cell cavity disappeared. As can be seen from Figure 4c,d, the vascular bundle tissue in the DIB and DHIB was obviously damaged, the fibers were broken, and the cell wall of the thin-walled cells was deformed. Furthermore, the cell wall was damaged. From Figure 4e,f, we can see that the TIB and THIB were covered with epoxy resin. There was no epoxy resin attached to the hollow part of the vascular bundle in the TIB. Compared with the THIB, the resin adhesion was uneven. 

From Figure 5, it can be seen that the OB sample contained complete vascular bundle tissue and parenchyma cell tissue. Compared with the IB, the wall of the parenchyma cell tissue in OB was thicker, and the cell cavity did not contain starch. Cells in both the DOB and DHOB were deformed and the vascular bundle tissue was destroyed. Meanwhile, cells in DHOB were more deformed and the degree of vascular tissue destruction was higher. The surface of TOB and THOB were covered with epoxy resin, and both were in a relatively uniform state.

The vascular bundle tissue and parenchyma cell tissues are presented in the IB and OB samples. The cell wall of the IB was thinner than OB, and the cell cavity of IB contained substances such as starch. After heat treatment, the starch material contained in IB was removed, and the cell wall thickened. The samples after delignification treatment showed vascular bundle tissue destruction and thin-walled cell deformation. Among them, the DOB vascular bundle tissue destruction was lowest, and fiber breakage was not obvious. The cross-section of the TB sample was covered with epoxy resin, indicating that the resin was successfully impregnated and cured in the cell lumen of the samples.

### 3.3. Lignin Content Testing

Bamboo is mainly composed of cellulose, hemicellulose, and lignin. Among them, cellulose and hemicellulose are colorless substances with a simple structure. Relatively speaking, the structure of lignin is relatively complicated [28], which is also one of the main factors for bamboo coloration [29].

The lignin content in the samples treated by the above delignification process was significantly reduced, as shown in Figure 6. This is because the sodium chlorite–water solution successfully entered the pores of the bamboo material under an acidic heating environment, the colored matter inside the bamboo material was largely removed, and the sample after delignification became colorless.

### 3.4. Light Transmittance Testing

Figure 7 shows that the original bamboo material and the heat-treated sample had no light transmittance, and the light transmittance of the pure epoxy resin (E51) was between 35–50%. Compared with the original bamboo sample, the transmittance of the transparent bamboo sample obtained after vacuum impregnation was therefore increased. Among them, the transmittance of the TIB sample could reach more than 10%, and the transmittance of the THIB sample was slightly smaller than that of the TIB sample, while the transmittance of the TOB sample was about 2%, and the transmittance of the THOB sample was slightly larger than that of the TOB sample. That is, the transmittance was TIB > THIB > THOB > TOB, and the transmittance increased with the increase in wavelength.

The transmission of the TIB sample was greater than that of the TOB sample. The heat treatment could decrease the transmittance of the IB sample, and increase the transmittance of the OB sample. From the comparison of Figure 4(e1,f1), it can be seen that the resin morphology on the TIB cross-sectional surface was more obvious than that of the THIB sample. From the comparison of Figure 5(e1,f1), it can be seen that the resin morphology on the TOB cross-sectional surface was less obvious than the THOB cross section. From a structural point of view, after the heat treatment, the resin content in THIB was less than the TIB, and the resin content in the THOB was greater than that of the TOB. Figure 8 shows the photos of the different types of bamboo and transparent bamboo samples, and the dimensions are listed in Table 1.

### 3.5. Mechanical Strength Testing

Table 2 presents the maximum tensile strength of the various samples tested in this test, which is an average value obtained by testing three parallel samples. The values in parentheses are the standard deviation (SD).

According to Figure 9, the maximum tensile strength of the heat-treated bamboo samples (HIB and HOB) were comparable to that of the control ones (IB and OB), indicating that the heat treatment did not significantly influence the tensile property of samples. The maximum tensile strength of the OB sample is significantly larger than that of the IB, indicating that the OB had higher fiber strength. This conclusion is consistent with Xian’s conclusion that the tensile strength of OB in Moso bamboo is greater than IB [30]. After the delignification treatment, the maximum tensile strength values of the DIB, DOB, DHIB, and DHOB samples were significantly decreased, which proved that the delignification treatment destroyed the internal fibers of the sample (also verified by the SEM images in Figure 4 and Figure 5) and weakened the tensile strength of the samples. The tensile strength of the TB samples increased, particularly the tensile strength of the TIB and TCOIB. It can be seen from Figure 9, that the maximum tensile strength values of the TIB and THIB samples were much greater than those of the delignified bamboo samples, which were similar to the values of the raw bamboo material samples. However, the maximum tensile strength values of the TOB and THOB samples were only slightly larger than those of the delignified bamboo template samples, and there was no distinct change. Compared with the raw bamboo material samples, the maximum tensile strength values of the delignified and TB samples both decreased significantly.

## 4. Conclusions

The transparent bamboo material was successfully prepared by the process adopted in this research. It could be seen from the SEM analysis that the vascular bundle tissue of the delignified bamboo template was broken and the thin-walled cells were deformed. The epoxy resin was attached to the cross-section of the TB samples after vacuum impregnation. This indicated that the epoxy resin was successfully impregnated into the pores of the bamboo material, and the internal structure of the transparent bamboo material was different from that of the original bamboo. The transmittance of transparent bamboo could reach up to about 15%. The transmittance of different parts of bamboo was different. In the visible light irradiation range, the transmittance of transparent bamboo materials was TIB > THIB and THOB > TOB. The tensile strength of the delignification bamboo was reduced compared to the raw bamboo material sample, which was due to the large amount of lignin being extracted and the damaged bamboo structure. The tensile strengths of different parts of transparent bamboo were different, among which the tensile strength of the TIB sample was not significantly lower than that of the IB, however, the TOB sample was significantly lower than that of the OB. From the perspective of mechanical properties, further research will be carried out to obtain a better preparation process to improve the mechanical properties of different parts of TB materials. Overall, this TB material is still restricted by mechanical strength, but it has good development prospect and can be used as decorative materials in the field of home and buildings.

## Figures and Tables

**Figure 1 polymers-12-00863-f001:**
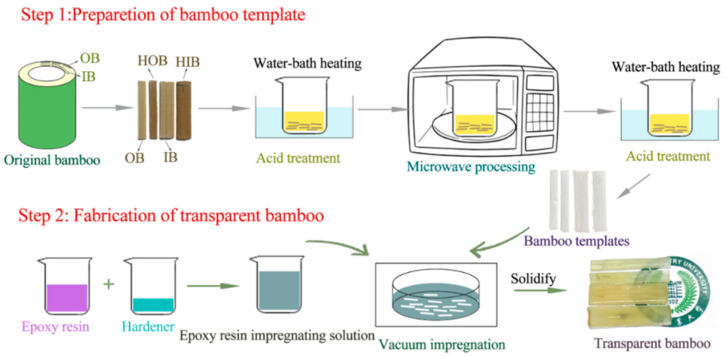
The main preparation process of transparent bamboo (TB).

**Figure 2 polymers-12-00863-f002:**
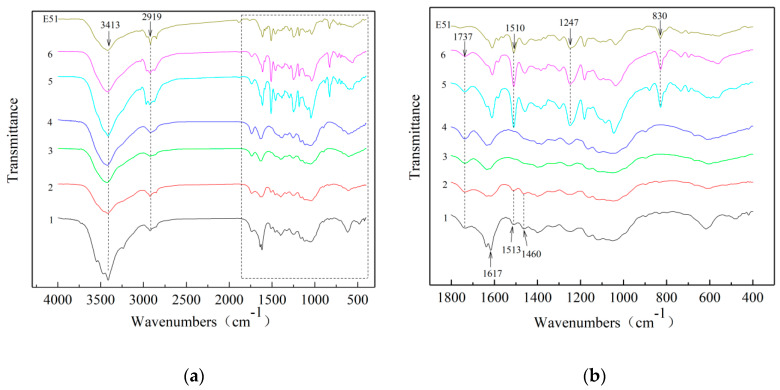
Fourier transform infrared (FTIR) curves of inner bamboo (IB) and heat-treated inner bamboo (HIB) before and after delignification and vacuum impregnation (**b**) is an enlarged view of the dotted box part of (**a**). (1) IB; (2) HIB; (3) DIB; (4) DHIB; (5) TIB; (6) THIB.

**Figure 3 polymers-12-00863-f003:**
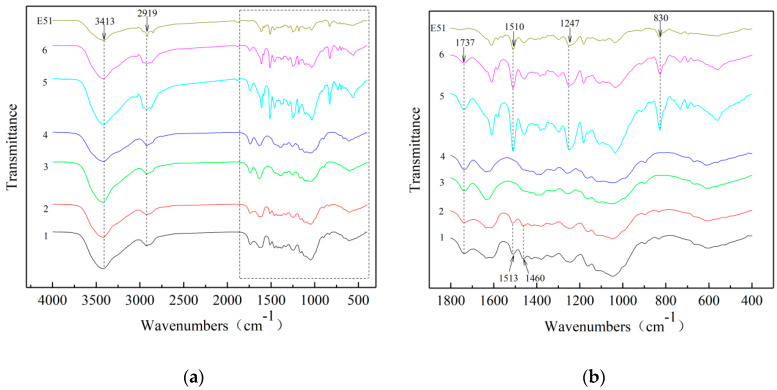
FTIR curves of outer bamboo (OB) and heat-treated outer bamboo (HOB) before and after delignification and vacuum impregnation ((**b**) is an enlarged view of the dotted box part of (**a**). (1) OB; (2) HOB; (3) delignified outer bamboo (DOB); (4) delignified heat-treated outer bamboo (DHOB); (5) transparent outer bamboo (TOB); (6) transparent heat-treated outer bamboo (THOB)).

**Figure 4 polymers-12-00863-f004:**
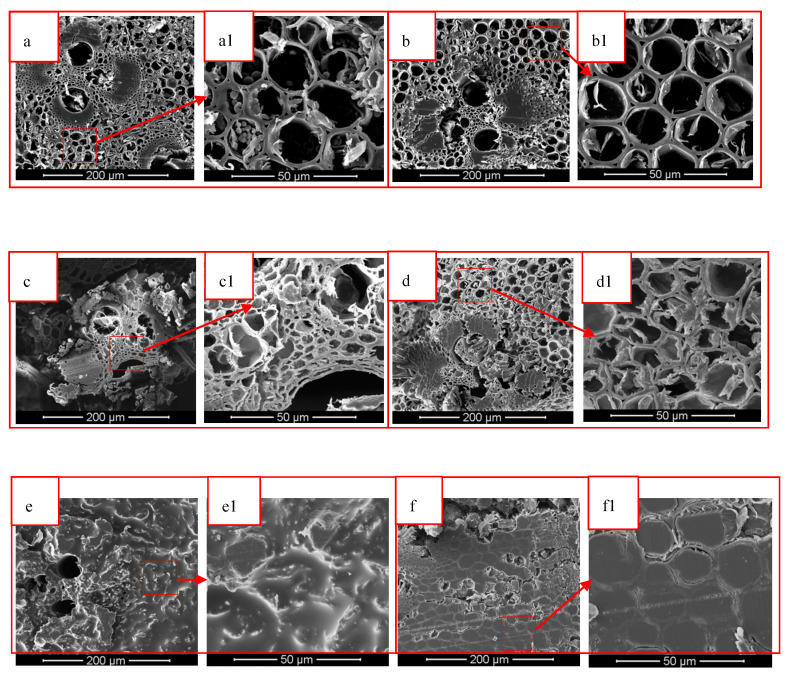
SEM images of IB and HIB before and after delignification and vacuum impregnation. ((**a**) IB; (**b**) HIB; (**c**) DIB; (**d**) DHIB; (**e**) TIB; (**f**) THIB; (**a1**–**f1**) are the enlarged images)).

**Figure 5 polymers-12-00863-f005:**
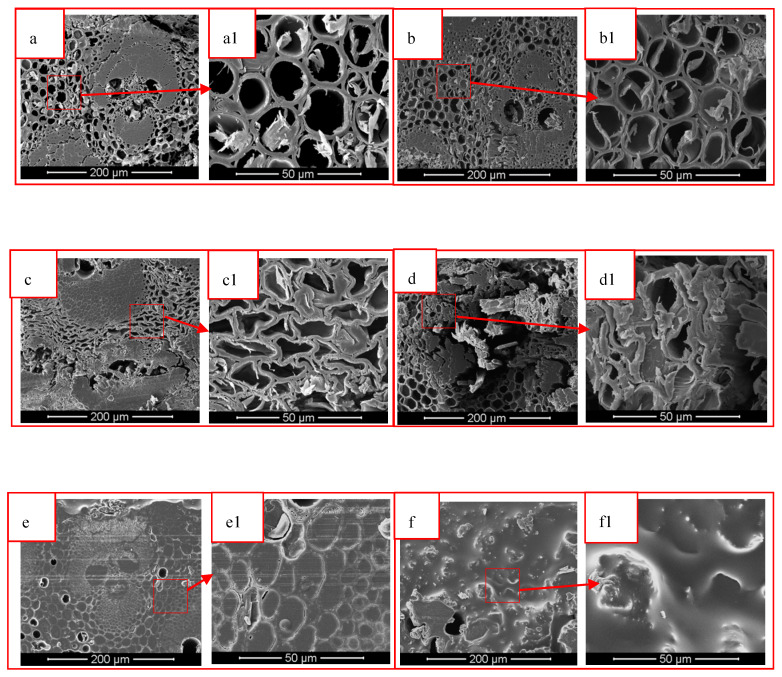
SEM images of OB and HOB before and after delignification and vacuum impregnation ((**a**) OB (**b**) HOB (**c**) DOB (**d**) DHOB (**e**) TOB (**f**) THOB; (**a1**–**f1**) were the enlarged images).

**Figure 6 polymers-12-00863-f006:**
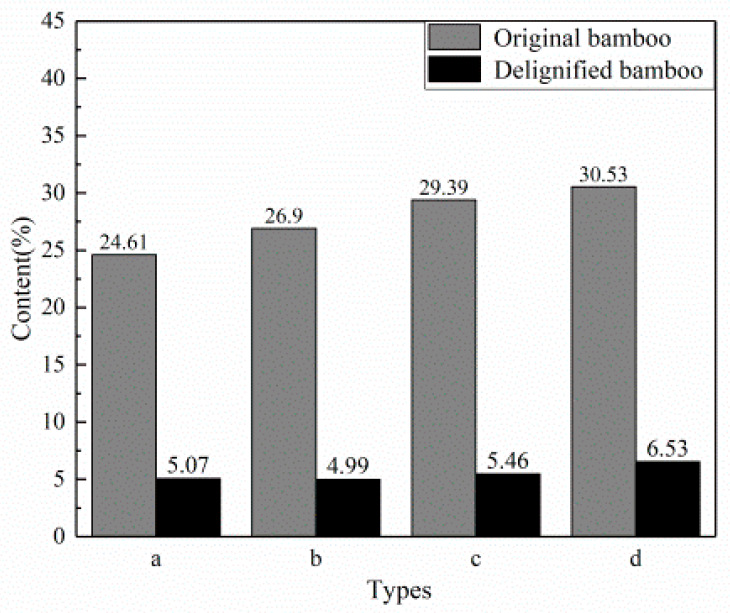
Lignin content of the delignified samples (a) IB; (b) HIB; (c) OB; (d) HOB.

**Figure 7 polymers-12-00863-f007:**
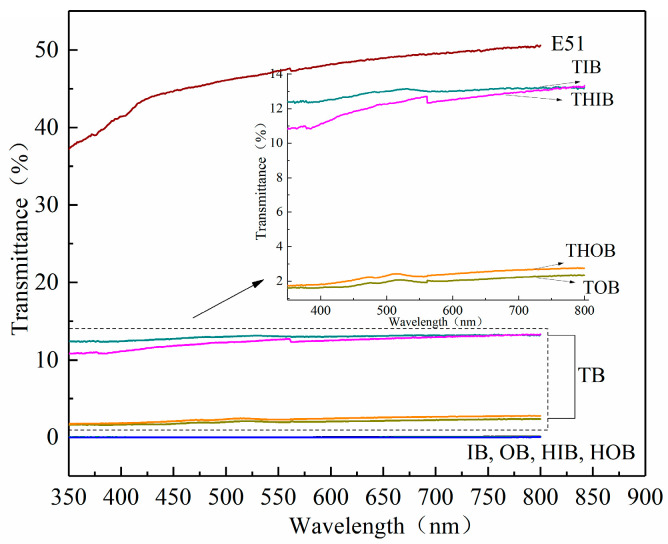
Comparison of the transmittance of different types of bamboo material and TB.

**Figure 8 polymers-12-00863-f008:**
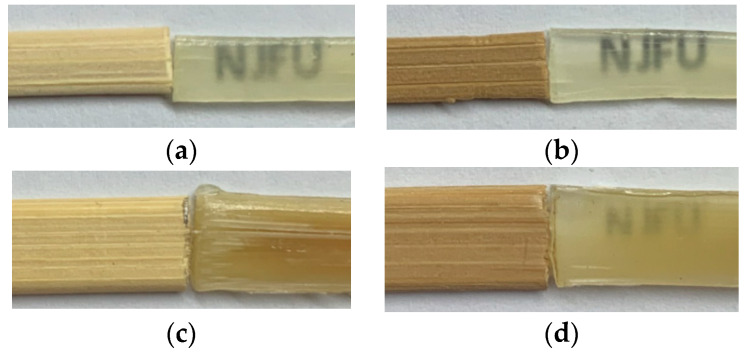
Photos of different types of bamboo material (left) and transparent bamboo (right). (**a**) IB and TIB; (**b**) HIB and THIB; (**c**) OB and TOB; (**d**) HOB and THOB).

**Figure 9 polymers-12-00863-f009:**
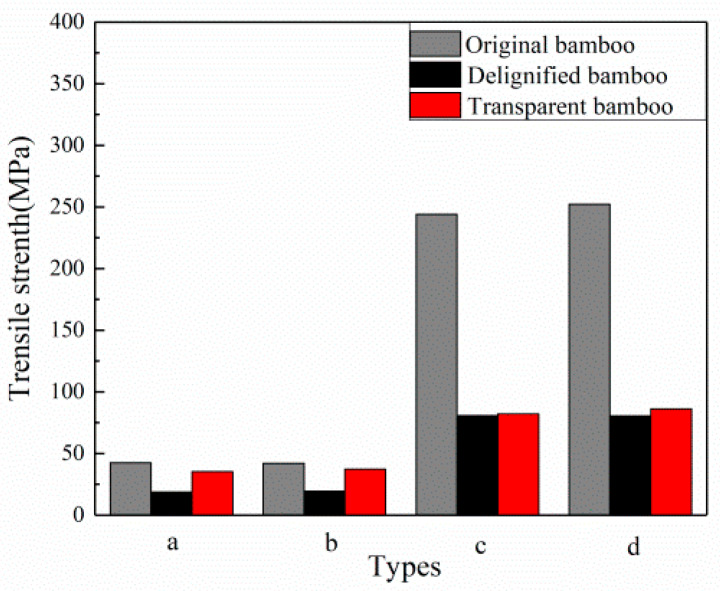
Comparison of the maximum tensile strength values in different samples. (a) IB; (b) HIB; (c) OB; (d) HOB.

**Table 1 polymers-12-00863-t001:** Specification of raw materials and the heat-treated materials.

Types	L (mm)	b (mm)	h (mm)	M_1_ (g)	M_2_ (g)	Absolute Dry Density (g·cm^−3^)
IB	3	4.40	1.1 ± 0.05	0.068 (0.001)	0.062 (0.002)	0.44
OB	3	7.80	1.8 ± 0.05	0.280 (0.018)	0.253 (0.014)	0.62
HIB	3	4.40	1.1 ± 0.05	0.065 (0.004)	0.062 (0.006)	0.40
HOB	3	7.80	1.8 ± 0.05	0.260 (0.013)	0.250 (0.010)	0.55

**Table 2 polymers-12-00863-t002:** Maximum tensile strength of the different samples.

Type		Maximum Tensile Strength (MPa)	Tensile Strength in Literature [30] (MPa)
**a**	IB	42.39 (3.95)	52.40
	DIB	18.71 (4.01)	--
	TIB	35.31 (3.78)	--
b	HIB	42.13 (1.29)	--
	DHIB	19.32 (5.13)	--
	THIB	37.28 (5.41)	--
c	OB	244.16 (6.94)	293.00
	DOB	80.70 (8.92)	--
	TOB	82.18 (3.81)	--
d	HOB	252.30 (16.29)	--
	DHOB	80.59 (12.46)	--
	THOB	86.17 (3.29)	--

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
