# Peer review of "Study on the Properties of Transparent Bamboo Prepared by Epoxy Resin Impregnation"

_polymers, 2020, doi:10.3390/polym12040863_

Round 1

Reviewer 1 Report

please see my comments in the attached file.

Author Response

Dear Editor and Reviewers: I am pleased to resubmit for the revised version of manuscript entitled “Study on the Properties of Transparent Bamboo Prepared by Epoxy Resin Impregnation” (ID: polymers-761943). Thank you for reading our manuscript and reviewing it. Those comments are all valuable and very helpful for revising and improving our paper. We have revised our manuscript carefully and have made correction which we hope meet with approval. So, we have sent the revised manuscript and have highlighted changes by using the yellow color. The main corrections in the paper and the responds to the reviewers’ comments are as following: Responds to the reviewers’ comments: Reviewer 1 Comments: 1) Line 14: omit the term ‘the’ Moso. Answer: It have already been deleted. 2) Line 15: omit the term ‘the’ raw materials and the term ‘for the’ transparent. Answer: They have already been deleted. 3) Add more keywords. Answer: I have added "heat treatment" and "acid delignification" as keywords in this paper. 4) Line 59: higher instead of larger. Give numbers (percentages) for this statement. Answer: The expression have been changed to "higher", and the proportion of the three elements is more than 80%. The expression in the paper was modified to "the higher chemical components in bamboo are cellulose, hemicellulose, and lignin, accounting for more than 80% [11]. Among them,..." 5)Before line 88, please specify the aim of the present study, in a paragraph. Answer: The research purpose of this article have been added, specifically stated in the paper is " At present, the research of transparent wood has made some progress, and its good optical properties make it become a new material to replace glass and other compounds in the field of architecture and home. Bamboo is also an excellent natural material, which has more obvious environmental advantages and mechanical properties than wood. Therefore, we consider to improve the light transmittance of bamboo to prepare transparent bamboo, so as to better replace the application of glass in the industrial field. " 6)Line 99-100: specify the equipment used and the supplier. Answer: Equipment and supplier have been added in the paper. Moso bamboo was produced in Anji, Zhejiang, China. The equipment used for the heat treatment was an electric constant temperature air-blast drying oven, which was provided by Xinmiao Medical Equipment Manufacturing Co., Ltd. Shanghai, China. 7)Line 104, Table 1: in the heading ‘number’ I suppose you mean the number or replicates used. If so, add the standard deviation in the columns Answer: Yes, the number in the table means the number of replicates used. Standard deviation has been added in columns M1 and M2. 8)Lines 112-114: give this information in Table 1. Answer: This information was added in Table 1. 9)Session 3.1: it is a well written and discussed session. Just, enlarge figures 2 & 3. Answer: I have enlarged the picture and inserted it into the text. 10)Session 3.3: I suggest you to perform a statistical analysis in the data shown in fig. 6 to identify significant differences. Also, improve the quality of fig. 6. Answer: As mentioned in 2.3.3, NREL method is used for lignin content test. In this paper, two groups of parallel samples are used for test results. I've changed Figure 6 to the picture shown below to compare the lignin content before and after delignification more clearly. 11)Session 3.4: it is a well written and discussed session. Just, enlarge fig. 7. Answer: I have enlarged the picture and inserted it into the text. 12)Line 296: what do you mean by the term statistical Table? I suggest you to perform a statistical analysis in the data Answer: Sorry, I didn't figure out the meaning of the statistics table. The data in Table 2 were obtained based on three parallel samples. The main ideal is that the tensile strength of the samples in different parts after different treatments is different. Therefore, the wording of statistical tables has been removed from the paper. 13)Line 300: remove it and give this info in the heading of the Table 2. Answer: It have been removed already. We appreciate for Editor and Reviewers’ warm work earnestly, and hope that the correction will meet with approval. Once again, thank you very much for your comments and suggestions. Yours sincerely, Corresponding author: Name: Yan Wu; Feng Yang E-mail: wuyan@njfu.edu.cn (Y.W.); yangfeng@bift.edu.cn (F.Y.)

Reviewer 2 Report

In the attached file there are many suggestions of improving the paper. English is very bad.

Improve figures as mentioned.

When talking about a figure or table in which you present results use present tense. When you talk about test you did use past tense.

Mechanical testing is not well explained and is not credible.

Author Response

I am pleased to resubmit for the revised version of manuscript entitled “Study on the Properties of Transparent Bamboo Prepared by Epoxy Resin Impregnation” (ID: polymers-761943). Thank you for reading our manuscript and reviewing it. Those comments are all valuable and very helpful for revising and improving our paper. We have revised our manuscript carefully and have made correction which we hope meet with approval. So, we have sent the revised manuscript and have highlighted changes by using the yellow color. The main corrections in the paper and the responds to the reviewers’ comments are as following:

Responds to the reviewers’ comments:

Reviewer 2 Comments: 

1) In the attached file there are many suggestions of improving the paper.

Improve figures as mentioned.

Answer: I have modified the paper based on the suggestions in the attachment.

2) When talking about a figure or table in which you present results use present tense. When you talk about test you did use past tense.

Answer: I have already modified.

3) Mechanical testing is not well explained and is not credible.

Answer: I have modified the expression, and the specific explanation is 10)-13) in the details in the attachment.

Details in the attachment

1) Table.1: write in header b and h ;write M1 and M2 in the table header.

Answer: Ok, I had changed the header, and the meaning of the abbreviated letters in the header was written below.

Table 1. Specification of raw materials and the heat-treated materials

Types

L (mm)

B (mm)

H (mm)

M1 (g)

M2 (g)

Absolute dry density (g·cm-3)

IB

3

4.40

1.1±0.05

0.068 (0.001)

0.062 (0.002)

0.44

OB

3

7.80

1.8±0.05

0.280 (0.018)

0.253 (0.014)

0.62

HIB

3

4.40

1.1±0.05

0.065 (0.004)

0.062 (0.006)

0.40

HOB

3

7.80

1.8±0.05

0.260 (0.013)

0.250 (0.010)

0.55

In the above table, L represents length, b represents width , h represents thickness, and SD represents standard deviation.

2) Line 143: be more precise; 12 or 24 h?

Answer: The curing time was at least 12 h, so I changed the expression to "The impregnated samples were then placed in the silica gel sheet and cured at room temperature for more than 12 h"

3) Line 169: describe samples; have they been done according to a standard?

Answer: TB samples are transparent bamboo samples, and the preparation method is the experimental method in 2.2. It can be obtained by curing after vacuum impregnating epoxy resin. In order to make readers understand more clearly, I had added the word “TB” in 2.2.3, specifically described as "The impregnated samples were then placed in the silica gel sheet and cured at room temperature for more than 12 h to obtain TB samples."

4) Line 171: what means normal? 

Answer: Normal temperature and pressure means room temperature and atmospheric pressure, and the expression had been modified in the paper.

5) Line 173: broke where? do you use an extensometer?

useless equations

Answer: The longitudinal fiber in the middle of the samples was broken. I didn't use the extensometer. I just put the test piece into the instrument mentioned in the paper for testing, and the computer can get the data by connecting the TRACEZIUM2 software matched with the instrument.

The useless equations were deleted.

6) Figure 2 and Figure 3: notate the two figures with a) and b) and explain which is which.

Answer: I have added the words a and b to the top left corner of each picture and added "Fig. b is an enlarged view of the dotted box part of Fig. a." to the picture title

7) Figure 4 and Figure 5: enlarge photos; scale cannot be seen

Answer: There are many pictures. If I enlarge them all, it will take a lot of space, so the magnification was marked in red on the lower right, so that the reader can clearly see the scale.

8) Line 276: who is larger?

Answer: The transmittance of the THOB sample is larger

9) Figure 7: enlarge diagram as to see writing; use better contrast colors for lines.

Answer: I have enlarged the words in the picture and changed the line color. The modified picture is as follows.

10) Line 296: what means "statistical table"?

Answer: Sorry, I didn't figure out the meaning of the statistics table. The data in Table 2 were obtained based on three parallel samples. The main ideal is that the tensile strength of the samples in different parts after different treatments is different. Therefore, the wording of statistical tables has been removed from the paper.

11) Table 2: average out of three tests? it is not relevant, especially without extensometer!

Answer: Average means that the data in this table is obtained after three parallel samples are subjected to tensile testing, mainly to show the accuracy of the data. Considering this confusion, I have changed the average to maximum. The data in the table are written for two reasons:1) It is compared with the existing literature, indicating that the maximum tensile force that OB can withstand is greater than the maximum tensile force that IB can withstand. The conclusion is consistent. 2) It is to provide data for plotting Figure 9.

12)Line 302: check text.

Answer: I have checked the text and modified this sentence to" the maximum tensile strength of the heat-treated bamboo samples (HIB and HOB) are comparable of the control ones (IB and OB), indicating that the heat treatment had not significantly influence on the tensile property of samples..."

13)Line 331-336: check text.

Answer: I have checked the text and modified this sentence to" The tensile strength of different parts of transparent bamboo was different, among which the tensile strength of TIB sample is not significantly lower than that of IB, however, TOB sample is significantly lower than that of OB. From the perspective of mechanical properties, further research will be carried out to obtain better preparation process, so as to improve the mechanical properties of different parts of TB materials. Overall, this TB material is still restricted by mechanical strength, but it has a good development prospect, and can be used as decorative materials in the field of home and building."

We appreciate for Editor and Reviewers’ warm work earnestly, and hope that the correction will meet with approval. Once again, thank you very much for your comments and suggestions.

Yours sincerely,

Name: Yan Wu; Feng Yang

E-mail: wuyan@njfu.edu.cn (Y.W.); yangfeng@bift.edu.cn (F.Y.)

Round 2

Reviewer 2 Report

In text all variables have to be written as italics (as L, b, h).